# Optimal 3D Angle of Arrival Sensor Placement with Gaussian Priors

**DOI:** 10.3390/e23111379

**Published:** 2021-10-21

**Authors:** Rongyan Zhou, Jianfeng Chen, Weijie Tan, Qingli Yan, Chang Cai

**Affiliations:** 1School of Marine Science and Technology, Northwestern Polytechnical University, Xi’an 710072, China or zhoury619@163.com (R.Z.); caichang@mail.nwpu.edu.cn (C.C.); 2School of Information Engineering, Nanyang Institute of Technology, Nanyang 473004, China; 3State Key Laboratory of Public Big Data, Guizhou University, Guiyang 550025, China; wjtan@gzu.edu.cn; 4School of Computer Science & Technology, Xi’an University of Posts & Telecommunications, Xi’an 710121, China; yql@xupt.edu.cn

**Keywords:** 3D angle of arrival (AOA) localization, Cramér–Rao lower bound (CRLB), optimal sensor placement, covariance matrix, fisher information matrix (FIM)

## Abstract

Sensor placement is an important factor that may significantly affect the localization performance of a sensor network. This paper investigates the sensor placement optimization problem in three-dimensional (3D) space for angle of arrival (AOA) target localization with Gaussian priors. We first show that under the A-optimality criterion, the optimization problem can be transferred to be a diagonalizing process on the AOA-based Fisher information matrix (FIM). Secondly, we prove that the FIM follows the invariance property of the 3D rotation, and the Gaussian covariance matrix of the FIM can be diagonalized via 3D rotation. Based on this finding, an optimal sensor placement method using 3D rotation was created for when prior information exists as to the target location. Finally, several simulations were carried out to demonstrate the effectiveness of the proposed method. Compared with the existing methods, the mean squared error (MSE) of the maximum a posteriori (MAP) estimation using the proposed method is lower by at least 25% when the number of sensors is between 3 and 6, while the estimation bias remains very close to zero (smaller than 0.15 m).

## 1. Introduction

Tracking and localization using sensor networks have a wide range of applications in radar, sonar, and wireless sensor networks [1,2]. There are several types of localization techniques that have been developed in recent years: time difference of arrival (TDOA) or time of arrival (TOA) [3,4], angle of arrival (AOA) [5,6,7], and received signal strength (RSS) [8,9].

AOA target localization has been an active research area during the past two decades. It does not require synchronization with the signal target or among the different distributed sensors, unlike TOA and TDOA localization. Many estimators have been developed for AOA-based localization. A 3D one-step pseudolinear estimator (PLE) with a bias compensation strategy was proposed in [10]. An asymptotically unbiased weight instrumental variable (WIV) technique was presented in [11] to solve the bias problem, and then a 3D, improved WIV estimator was derived to break down the correlation between the instrumental variable (IV) matrix and the error vector in [12]. Furthermore, a closed-form solution for 3D AOA localization, which can handle the presence of sensor location errors, was presented in [13]. Recently, an approximately unbiased estimator was proposed by approximating the bias and subtracting it from the weighted least squares (WLS) solution obtained using semidefinite relaxation (SDR) in [14].

Apart from the above localization methods, generating the target–sensor geometry for localization is also a non-trivial task and attracts great interest in the localization area. The optimization problem for sensor placement was usually formulated to minimize the Cramer–Rao lower bound (CRLB) or maximize the Fisher information matrix (FIM) [15,16,17,18], and the differences between the above two methods were reported in [19]. In [20], the trace of CRLB was adopted to find the optimal geometric configuration, which yielded the minimum possible covariance of any unbiased target estimator in a constrained 3D space. The optimal placement analysis for 3D AOA target localization using the A-optimality criterion (minimize the trace of CRLB) appeared in [21]. In addition, a frame theory was also presented that can handle the optimal sensor placement with three types of sensor placement strategy in [22] as an identical parameter optimization problem in two-dimensional (2D) and 3D space. In [23], the frame theory was used to derive an evaluation function for optimal placement with random numbers of newly added sensors in AOA target localization.

The majority of previous work on optimal sensor placement assumed that the target location was known perfectly, which is impossible in actual scenarios. Therefore, it is beneficial to solve the optimal sensor placement problem when the target location is uncertain. The optimal sensor placement algorithm for TDOA localization with an unknown target location was proposed in [24]. An equivalence between minimizing the estimation mean squared error and minimizing the area of the estimation uncertainty ellipse was established for the geometry optimization problem of target localization with Bayesian priors in [25], which makes the optimal geometry conditions algebraically simple and easy to be computed. However, the above proposed algorithms can only be used in 2D space. In addition, an analysis of the performance measures of covariance and information matrices in resource management for target state estimation was provided in [26]. Then the analysis results were extended in [27] to find the optimal placement of heterogeneous sensors for the target with Gaussian priors. Furthermore, the updated FIM was used to derive optimal placement conditions for heterogeneous sensors tracking the unknown number of targets in [28]. Nevertheless, the solutions in [27,28] were complicated, particularly in the case of more than two sensors.

Several valuable conclusions have been obtained about the coordinate system rotation, which provides a new path to solving target localization and optimal sensor placement. As pointed out in [29], local coordinate translations and rotations do not influence the PLE and maximum likelihood estimator (MLE) performance of the bearings-only target localization algorithm. Furthermore, it was demonstrated that the trace of CRLB was invariant in XY-coordinates and the AOA-based FIM was invariant to flipping a sensor about the target in [21]. Lately, a TOA-based FIM invariant to sensor rotation about the target in 3D space was shown in [30].

In this paper, we address the optimal 3D AOA sensor placement problem with Gaussian priors. The key contributions of this paper are summarized as follows:A detailed 3D AOA optimal sensor placement problem with Gaussian priors is analyzed using the A-optimality criterion (minimizing the trace of the inverse FIM). We show analytically that the problem can be transformed to diagonalize the AOA-based FIM under the A-optimality criterion.The invariance property of the 3D rotation for the AOA-based FIM with Gaussian priors is deduced. Thus, the Gaussian covariance matrix of the FIM can be diagonalized via 3D rotation.An optimal sensor placement method using 3D rotation is proposed for when prior information exists as to the target location using the invariance property of the AOA-based FIM and the A-optimality criterion.Simulation studies are presented to demonstrate the analytical findings. The comparison results show that the proposed method significantly improves the localization performance.

The rest of the paper is organized as follows: The 3D AOA sensor placement with Gaussian priors optimization problem is formulated in Section 2. Section 3 derives the FIM with Gaussian priors after the 3D rotation and then exploits the invariance property for the 3D AOA-based FIM. Section 4 presents the optimal sensor-target geometric solutions with the help of a resistor network analogy. The main results are presented with simulation examples in Section 5, and the conclusion and discussion of future work are in Section 6.

## 2. Problem Formulation

We consider a 3D AOA configuration with *N* sensors localizing a stationary target, as depicted in Figure 1, and each sensor is assumed to be omnidirectional. $s={\left(x,y,z\right)}^{T}$ is the unknown location of the target with ${}^{T}$ denoting matrix transpose, $p_{k}\;=\;{\left(p_{xk},p_{yk},p_{zk}\right)}^{T},k\;=\;1,2,\ldots N$ is the location of the sensors. It is assumed that s is a Gaussian random variable with a distribution as $s∼N\left(s_{0},P_{0}\right)$, where $s_{0}$ and $P_{0}$ represent the mean and the covariance matrix of s. Note that the gray ellipse in Figure 1 illustrates the confidence region corresponding to the Gaussian priors, and $\left\{\theta_{k},\phi_{k}\right\}$ denotes the bearing measurement with the azimuth and elevation angle in spherical coordinates. Using $s_{0}={\left(x_{0},y_{0},z_{0}\right)}^{T}$ as a reference, the AOA measurement of the *k*th sensor can be expressed as
(1)$$\begin{array}{c} \begin{array}{cc} \; & \theta_{k}=\tan^{-1}\frac{y_{0}-p_{yk}}{x_{0}-p_{xk}},-\pi <\theta \leq \pi ,\\ \; & \phi_{k}=\sin^{-1}\frac{z_{0}-p_{zk}}{r_{k}},-\frac{\pi }{2}<\phi \leq \frac{\pi }{2}, \end{array} \end{array}$$
where $r_{k}=∥s_{0}-p_{k}∥$, $\tan^{-1}$ is the fourth quadrant arctangent, and $∥\cdot ∥$ denotes the Euclidean norm. In terms of azimuth and elevation angles, the unit bearing vector $g_{k}^{0}$ can be given by
(2)$$\begin{array}{c} \begin{array}{c} g_{k}^{0}=\left[\begin{array}{c} \cos\phi_{k}\cos\theta_{k}\\ \cos\phi_{k}\sin\theta_{k}\\ \sin\phi_{k} \end{array}\right], \end{array} \end{array}$$

In the 3D localization system, the AOA measurements are always affected by multipath effects, the propagation environment, the transmitted power, and other unfavorable factors. In order to focus our study on the sensor placement optimization problem itself, in our paper, although we do not consider these inference factors explicitly, we take them into account, as a whole, by modeling them as the additive Gaussian white noise on the true angle measurements $\left\{{\tilde{\theta }}_{k},{\tilde{\phi }}_{k}\right\}$ as
(3)$$\begin{array}{c} \begin{array}{cc} \; & {\tilde{\theta }}_{k}=\theta_{k}+n_{\theta_{k}},n_{\theta_{k}}∼N\left(0,\sigma_{\theta_{k}}^{2}\right),\\ \; & {\tilde{\phi }}_{k}=\phi_{k}+n_{\phi_{k}},n_{\phi_{k}}∼N\left(0,\sigma_{\phi_{k}}^{2}\right). \end{array} \end{array}$$
where $\sigma_{\theta_{k}}^{2}$ and $\sigma_{\phi_{k}}^{2}$ are sensor-dependent noise variances [31].

The sensor measurement covariance matrix can be expressed as
(4)$$\begin{array}{c} \begin{array}{c} \Sigma =\left[\begin{array}{cc} P_{0} & 0_{2N\times 3}\\ 0_{3\times 2N} & \Sigma_{0} \end{array}\right], \end{array} \end{array}$$
with
(5)$$\begin{array}{c} \begin{array}{c} \Sigma_{0}=\mathrm{diag}\left\{\sigma_{\theta_{1}}^{2},\sigma_{\phi_{1}}^{2},\ldots ,\sigma_{\theta_{N}}^{2},\sigma_{\phi_{N}}^{2}\right\}, \end{array} \end{array}$$

Here we define $e\left(s\right)$ and $r\left(s\right)$
(6)$$\begin{array}{c} \begin{array}{cc} \; & r\left(s\right)=s-s_{0},\\ \; & e\left(s\right)=\left[{\tilde{\theta }}_{1}-\theta_{1}\left(s\right),{\tilde{\phi }}_{1}-\phi_{1}\left(s\right),\right.{\left.\ldots ,{\tilde{\theta }}_{N}-\theta_{N}\left(s\right),{\tilde{\phi }}_{N}-\phi_{N}\left(s\right)\right]}^{T}. \end{array} \end{array}$$

The Jacobian matrix of measurement errors evaluated at the mean location $s_{0}$ can be written as
(7)$$\begin{array}{c} \begin{array}{c} J={\left[\begin{array}{cc} J_{1} & J_{2} \end{array}\right]}^{T}, \end{array} \end{array}$$
where $J_{1}$ is the 3×3 Jacobian of $r\left(s\right)$, given by
(8)$$\begin{array}{c} \begin{array}{cc} \; & J_{1}=I_{3\times 3} \end{array} \end{array},$$

The Jacobian vector of the kth sensor measurement error evaluated at the true target location $s={\left(x,y,z\right)}^{T}$ as
(9)$$\begin{array}{c} \begin{array}{cc} J_{k}^{{}^{\prime }}\; & ={\left.{\left[\frac{\partial \theta_{k}}{\partial s^{T}},\frac{\partial \phi_{k}}{\partial s^{T}}\right]}^{T}\right|}_{s}={\left.\left[\begin{array}{c} \begin{array}{ccc} \frac{\partial \theta_{k}}{\partial x} & \frac{\partial \theta_{k}}{\partial y} & \frac{\partial \theta_{k}}{\partial z} \end{array}\\ \begin{array}{ccc} \frac{\partial \phi_{k}}{\partial x} & \frac{\partial \phi_{k}}{\partial y} & \frac{\partial \phi_{k}}{\partial z} \end{array} \end{array}\right]\right|}_{s}\\ \; & =\left[\begin{array}{ccc} -\frac{\sin\theta_{k}}{r_{k}\cos\phi_{k}} & \frac{\cos\theta_{k}}{r_{k}\cos\phi_{k}} & 0\\ -\frac{\sin\phi_{k}\cos\theta_{k}}{r_{k}} & -\frac{\sin\phi_{k}\sin\theta_{k}}{r_{k}} & \frac{\cos\phi_{k}}{r_{k}} \end{array}\right], \end{array} \end{array}$$

Therefore, we can obtain the Jacobian matrix of the 2N measurements as
(10)$$\begin{array}{c} \begin{array}{cc} J_{2} & =\left[\begin{array}{ccc} -\frac{\sin\theta_{1}}{r_{1}\cos\phi_{1}} & \frac{\cos\theta_{1}}{r_{1}\cos\phi_{1}} & 0\\ -\frac{\sin\phi_{1}\cos\theta_{1}}{r_{1}} & -\frac{\sin\phi_{1}\sin\theta_{1}}{r_{1}} & \frac{\cos\phi_{1}}{r_{1}}\\ \vdots & \vdots & \vdots \\ -\frac{\sin\theta_{N}}{r_{N}\cos\phi_{N}} & \frac{\cos\theta_{N}}{r_{N}\cos\phi_{N}} & 0\\ -\frac{\sin\phi_{N}\cos\theta_{N}}{r_{N}} & -\frac{\sin\phi_{N}\sin\theta_{N}}{r_{N}} & \frac{\cos\phi_{N}}{r_{N}} \end{array}\right], \end{array} \end{array}$$

The FIM for 3D AOA localization with Gaussian problem yields
(11)$$\begin{array}{c} \begin{array}{c} \Phi =J^{T}\Sigma^{-1}J. \end{array} \end{array}$$

For simplification, J is expressed as the following three vectors:(12)$$\begin{array}{c} \begin{array}{cc} \; & a=\left[-\frac{\sin\theta_{1}}{r_{1}\cos\phi_{1}},-\frac{\sin\phi_{1}\cos\theta_{1}}{r_{1}},\cdots ,\right.{\left.-\frac{\sin\theta_{N}}{r_{N}\cos\phi_{N}},-\frac{\sin\phi_{N}\cos\theta_{N}}{r_{N}}\right]}^{T},\\ \; & b=\left[\frac{\cos\theta_{1}}{r_{1}\cos\phi_{1}},-\frac{\sin\phi_{1}\sin\theta_{1}}{r_{1}},\cdots ,\right.{\left.\frac{\cos\theta_{N}}{r_{N}\cos\phi_{N}},-\frac{\sin\phi_{N}\sin\theta_{N}}{r_{N}}\right]}^{T},\\ \; & c={\left[0,\frac{\cos\phi_{1}}{r_{1}},\cdots ,0,\frac{\cos\phi_{N}}{r_{N}}\right]}^{T}, \end{array} \end{array}$$

Thus,
(13)$$\begin{array}{c} \begin{array}{c} J={\left[\begin{array}{ccc} a & b & c \end{array}\right]}_{(2N+3)\times 3} \end{array} \end{array},$$

Hence, the FIM is
(14)$$\begin{array}{c} \begin{array}{c} \Phi =\left[\begin{array}{c} a^{T}\\ b^{T}\\ c^{T} \end{array}\right]\Sigma^{-1}\left[\begin{array}{ccc} a & b & c \end{array}\right]=\left[\begin{array}{ccc} {\hat{a}}^{T}\hat{a} & {\hat{a}}^{T}\hat{b} & {\hat{a}}^{T}\hat{c}\\ {\hat{b}}^{T}\hat{a} & {\hat{b}}^{T}\hat{b} & {\hat{b}}^{T}\hat{c}\\ {\hat{c}}^{T}\hat{a} & {\hat{c}}^{T}\hat{b} & {\hat{c}}^{T}\hat{c} \end{array}\right], \end{array} \end{array}$$
where $\hat{a}=\Sigma^{-1/2\;}a$, $\hat{b}=\Sigma^{-1/2\;}b$, $\hat{c}=\Sigma^{-1/2\;}c$, and $\Sigma^{-1/2\;}\Sigma^{-1/2\;}=\Sigma^{-1}$. Given $\hat{a}$, $\hat{b}$, and $\hat{c}$ in ${ℜ}^{2n}$, then ${\left|\hat{a}\right|}^{2}=\langle \hat{a},\hat{a}\rangle$ and $\langle \hat{a},\hat{b}\rangle =\left|\hat{a}\right|\left|\hat{b}\right|\cos\left(\theta_{\hat{a}\hat{b}}\right)$, from which it follows that the angle $\theta_{\hat{a}\hat{b}}$ between vector $\hat{a}$ and $\hat{b}$ is given by $\theta_{\hat{a}\hat{b}}=\cos^{-1}\left(\langle \hat{a},\hat{b}\rangle /\left(\left|\hat{a}\right|\left|\hat{b}\right|\right)\;\right)$, $\theta_{\hat{a}\hat{c}}$. $\theta_{\hat{b}\hat{c}}$ are the angle defined by vectors $\hat{a}$; and $\hat{c}$, $\hat{b}$, and $\hat{c}$ [32]. With this notion, the FIM becomes
(15)$$\begin{array}{c} \begin{array}{c} \Phi =\left[\begin{array}{ccc} {\left|\hat{a}\right|}^{2} & \left|\hat{a}\right|\left|\hat{b}\right|\cos\left(\theta_{\hat{a}\hat{b}}\right) & \left|\hat{a}\right|\left|\hat{c}\right|\cos\left(\theta_{\hat{a}\hat{c}}\right)\\ \left|\hat{a}\right|\left|\hat{b}\right|\cos\left(\theta_{\hat{a}\hat{b}}\right) & {\left|\hat{b}\right|}^{2} & \left|\hat{b}\right|\left|\hat{c}\right|\cos\left(\theta_{\hat{b}\hat{c}}\right)\\ \left|\hat{a}\right|\left|\hat{c}\right|\cos\left(\theta_{\hat{a}\hat{c}}\right) & \left|\hat{b}\right|\left|\hat{c}\right|\cos\left(\theta_{\hat{b}\hat{c}}\right) & {\left|\hat{c}\right|}^{2} \end{array}\right], \end{array} \end{array}$$

The determinant of Φ is
(16)$$\begin{array}{c} \begin{array}{c} \left|\Phi \right|={\left|\hat{a}\right|}^{2}{\left|\hat{b}\right|}^{2}{\left|\hat{c}\right|}^{2}\lambda , \end{array} \end{array}$$
where
(17)$$\begin{array}{c} \begin{array}{c} \lambda =1-\cos^{2}\left(\theta_{\hat{a}\hat{b}}\right)-\cos^{2}\left(\theta_{\hat{a}\hat{c}}\right)-\cos^{2}\left(\theta_{\hat{b}\hat{c}}\right)+2\cos\left(\theta_{\hat{a}\hat{b}}\right)\cos\left(\theta_{\hat{a}\hat{c}}\right)\cos\left(\theta_{\hat{b}\hat{c}}\right). \end{array} \end{array}$$

Thus, the trace of CRLB is
(18)$$\begin{array}{c} \begin{array}{cc} \; & \mathrm{tr}\left(\mathrm{CRLB}\right)=\mathrm{tr}\left(\Phi^{-1}\right)\\ \; & =\frac{{\left|\hat{b}\right|}^{2}{\left|\hat{c}\right|}^{2}\left(1-\cos^{2}\left(\theta_{\hat{b}\hat{c}}\right)\right)}{\left|\Phi \right|}+\frac{{\left|\hat{a}\right|}^{2}{\left|\hat{c}\right|}^{2}\left(1-\cos^{2}\left(\theta_{\hat{a}\hat{c}}\right)\right)}{\left|\Phi \right|}+\frac{{\left|\hat{a}\right|}^{2}{\left|\hat{b}\right|}^{2}\left(1-\cos^{2}\left(\theta_{\hat{a}\hat{b}}\right)\right)}{\left|\Phi \right|}\\ \; & =\frac{\left(1-\cos^{2}\left(\theta_{\hat{b}\hat{c}}\right)\right)}{{\left|\hat{a}\right|}^{2}\lambda }+\frac{\left(1-\cos^{2}\left(\theta_{\hat{a}\hat{c}}\right)\right)}{{\left|\hat{b}\right|}^{2}\lambda }+\frac{\left(1-\cos^{2}\left(\theta_{\hat{a}\hat{b}}\right)\right)}{{\left|\hat{c}\right|}^{2}\lambda }, \end{array} \end{array}$$

Thus, we can get
(19)$$\begin{array}{c} \begin{array}{c} \mathrm{tr}\left(\mathrm{CRLB}\right)\geq \frac{1}{{\left|\hat{a}\right|}^{2}}+\frac{1}{{\left|\hat{b}\right|}^{2}}+\frac{1}{{\left|\hat{c}\right|}^{2}}. \end{array} \end{array}$$

The $\mathrm{tr}\left(\mathrm{CRLB}\right)$ is minimum when $\cos\left(\theta_{\hat{a}\hat{b}}\right)=\cos\left(\theta_{\hat{a}\hat{c}}\right)=\cos\left(\theta_{\hat{b}\hat{c}}\right)=0$. Note that when $\mathrm{tr}\left(\mathrm{CRLB}\right)$ becomes minimum, the FIM becomes diagonal, so the optimal sensor placement is obtained by diagonalizing the FIM [33].

## 3. The Proposed Method

Under the Gaussian assumption, the prior covariance matrix $P_{0}$ may be a diagonal or non-diagonal matrix, which physically represents an ellipsoid bounding the uncertain target measurement estimators. Since the rotation does not affect the size of the ellipsoid, the covariance $P_{0}$ should be invariant to any similarity transform $UP_{0}U^{T}$, where U is a unitary matrix. Therefore, a proper 3D rotation provides a solution for diagonalizing the non-diagonal matrix $P_{0}$. Additionally, in this section, we derive the FIM for 3D AOA localization with Gaussian priors after the 3D rotation, and then the invariance property of the 3D AOA-based FIM is exploited.

### 3.1. 3D Rotation Matrix

First, we define rotation matrices of the AOA measurement as follows:
(20)$$\begin{array}{c} \begin{array}{cc} \; & R_{x}=\left(\begin{array}{ccc} 1 & 0 & 0\\ 0 & \cos\alpha & -\sin\alpha \\ 0 & \sin\alpha & \cos\alpha \end{array}\right),R_{y}=\left(\begin{array}{ccc} \cos\beta & 0 & \sin\beta \\ 0 & 1 & 0\\ -\sin\beta & 0 & \cos\beta \end{array}\right),R_{z}=\left(\begin{array}{ccc} \cos\gamma & -\sin\gamma & 0\\ \sin\gamma & \cos\gamma & 0\\ 0 & 0 & 1 \end{array}\right). \end{array} \end{array}$$

Here α, β, and γ are counterclockwise rotation angles around the *x*, *y*, and *z* axes, respectively, which is depicted in Figure 2. The rotation matrix is
(21)$$\begin{array}{c} \begin{array}{c} R=R_{x}R_{y}R_{z}. \end{array} \end{array}$$
and satisfies $RR^{T}=RR^{-1}=I$.

Next, when the rotation happens in the 3D space, we can get
(22)$$\begin{array}{c} \begin{array}{c} s^{r}=Rs,\;s_{0}^{r}=Rs_{0},\;p^{r}=Rp,\;P_{0}^{r}=RP_{0}R^{T}. \end{array} \end{array}$$
where $s^{r}$, $s_{0}^{r}$, $p^{r}$, and $P_{0}^{r}$ are the new measurements compared with s, $s_{0}$, p, and $P_{0}$ after rotation.

### 3.2. Invariance to 3D Rotation for AOA-Based FIM

When the 3D AOA measurements are assumed to be corrupted by additive white Gaussian noise with zero mean, the *k*-th sensor bearing unit vector in (Equation 2) is modified as
(23)$$\begin{array}{c} \begin{array}{c} g_{k}=\left[\begin{array}{c} \cos{\tilde{\phi }}_{k}\cos{\tilde{\theta }}_{k}\\ \cos{\tilde{\phi }}_{k}\sin{\tilde{\theta }}_{k}\\ \sin{\tilde{\phi }}_{k} \end{array}\right], \end{array} \end{array}$$

From (Equation 20) and (Equation 22), the bearing unit vector after rotation is
(24)$$\begin{array}{c} \begin{array}{c} g_{k}^{r}=Rg_{k}=\left[\begin{array}{c} \cos{\tilde{\phi }}_{k}^{r}\cos{\tilde{\theta }}_{k}^{r}\\ \cos{\tilde{\phi }}_{k}^{r}\sin{\tilde{\theta }}_{k}^{r}\\ \sin{\tilde{\phi }}_{k}^{r} \end{array}\right], \end{array} \end{array}$$

Therefore, the azimuth and elevation angles are given by
(25)$$\begin{array}{c} \begin{array}{c} {\tilde{\theta }}_{k}^{r}=\tan^{-1}\left(\frac{g_{k}^{r}\left(2\right)}{g_{k}^{r}\left(1\right)}\right),\;{\tilde{\phi }}_{k}^{r}=\sin^{-1}\left(g_{k}^{r}\left(3\right)\right). \end{array} \end{array}$$

Here we define
(26)$$\begin{array}{c} \begin{array}{c} {\tilde{\theta }}_{k}^{r}=g\left({\tilde{\theta }}_{k},{\tilde{\phi }}_{k}\right),\;{\tilde{\phi }}_{k}^{r}=h\left({\tilde{\theta }}_{k},{\tilde{\phi }}_{k}\right), \end{array} \end{array}$$

To compute the covariance matrix after rotation, we can adopt the First-order Taylor series approximation for the rotated noisy angles using $\left(\theta_{k},\phi_{k}\right)$ in $\left({\tilde{\theta }}_{k},{\tilde{\phi }}_{k}\right)$ with respect to the noise variables $n_{\theta k}$ and $n_{\phi k}$. Therefore, (Equation 26) can be rewritten as
(27)$$\begin{array}{c} \begin{array}{cc} {\tilde{\theta }}_{k}^{r} & =g\left({\tilde{\theta }}_{k},{\tilde{\phi }}_{k}\right)=g\left(\theta_{k}+n_{\theta_{k}},\phi_{k}+n_{\phi_{k}}\right)=g\left(\theta_{k},\phi_{k}\right)+\left[\begin{array}{cc} \frac{\partial g\left(\theta_{k},\phi_{k}\right)}{\partial \theta_{k}} & \frac{\partial g\left(\theta_{k},\phi_{k}\right)}{\partial \phi_{k}} \end{array}\right]\left[\begin{array}{c} n_{\theta_{k}}\\ n_{\phi_{k}} \end{array}\right],\\ {\tilde{\phi }}_{k}^{r} & =h\left({\tilde{\theta }}_{k},{\tilde{\phi }}_{k}\right)=h\left(\theta_{k}+n_{\theta_{k}},\phi_{k}+n_{\phi_{k}}\right)=h\left(\theta_{k},\phi_{k}\right)+\left[\begin{array}{cc} \frac{\partial h\left(\theta_{k},\phi_{k}\right)}{\partial \theta_{k}} & \frac{\partial h\left(\theta_{k},\phi_{k}\right)}{\partial \phi_{k}} \end{array}\right]\left[\begin{array}{c} n_{\theta_{k}}\\ n_{\phi_{k}} \end{array}\right]. \end{array} \end{array}$$

According to the error propagation law [34], the noise covariance matrix for the *k*-th sensor after 3D rotation can be written as
(28)$$\begin{array}{c} \begin{array}{cc} K_{k}^{r}= & \left[\begin{array}{cc} \frac{\partial g\left(\theta_{k},\phi_{k}\right)}{\partial \theta_{k}} & \frac{\partial g\left(\theta_{k},\phi_{k}\right)}{\partial \phi_{k}}\\ \frac{\partial h\left(\theta_{k},\phi_{k}\right)}{\partial \theta_{k}} & \frac{\partial h\left(\theta_{k},\phi_{k}\right)}{\partial \phi_{k}} \end{array}\right]\times \left[\begin{array}{cc} \sigma_{\theta }^{2} & 0\\ 0 & \sigma_{\phi }^{2} \end{array}\right]\times {\left[\begin{array}{cc} \frac{\partial g\left(\theta_{k},\phi_{k}\right)}{\partial \theta_{k}} & \frac{\partial g\left(\theta_{k},\phi_{k}\right)}{\partial \phi_{k}}\\ \frac{\partial h\left(\theta_{k},\phi_{k}\right)}{\partial \theta_{k}} & \frac{\partial h\left(\theta_{k},\phi_{k}\right)}{\partial \phi_{k}} \end{array}\right]}^{T}, \end{array} \end{array}$$

By substituting (Equation 26) into (Equation 27), the maximum likelihood (ML) covariance matrix of the bearing measurement noise can be expressed as
(29)$$\begin{array}{c} \begin{array}{c} \Sigma_{0}^{r}=\left[\begin{array}{ccc} K_{1}^{r} & 0 & 0\\ 0 & \ddots & 0\\ 0 & 0 & K_{N}^{r} \end{array}\right], \end{array} \end{array}$$

Using the prior covariance matrix after rotation $P_{0}^{r}$ given in (Equation 22) and the above equation, the covariance matrix after rotation is given by
(30)$$\begin{array}{c} \begin{array}{c} \Sigma^{r}=\left[\begin{array}{cc} P_{0}^{r} & 0_{2N\times 3}\\ 0_{3\times 2N} & \Sigma_{0}^{r} \end{array}\right]. \end{array} \end{array}$$

By substituting (Equation 22) into (Equation 8) and (Equation 10), $J_{1}^{r}$ and $J_{2}^{r}$ after rotation are computed. We thus obtain
(31)$$\begin{array}{c} \begin{array}{c} J^{r}={\left[\begin{array}{cc} J_{1}^{r} & J_{2}^{r} \end{array}\right]}^{T}, \end{array} \end{array}$$

Hence, the FIM after three rotations becomes
(32)$$\begin{array}{c} \begin{array}{c} \hat{\Phi }={J^{r}}^{T}{\left(\Sigma^{r}\right)}^{-1}J^{r}. \end{array} \end{array}$$

After the 3D rotations, the FIM becomes
(33)$$\begin{array}{c} \begin{array}{c} \hat{\Phi }=R\Phi R^{-1}, \end{array} \end{array}$$

Substituting (Equation 21) into the above equation yields
(34)$$\begin{array}{c} \begin{array}{c} \hat{\Phi }=R_{x}R_{y}R_{z}\Phi {R_{z}}^{-1}{R_{y}}^{-1}{R_{x}}^{-1}, \end{array} \end{array}$$

By using ${\left(\mathrm{AB}\right)}^{-1}=B^{-1}A^{-1}$, A and B are full rank square matrices. The inverse of the new FIM $\hat{\Phi }$ is
(35)$$\begin{array}{c} \begin{array}{c} {\hat{\Phi }}^{-1}=R_{x}R_{y}R_{z}\Phi^{-1}{R_{z}}^{-1}{R_{y}}^{-1}{R_{x}}^{-1}, \end{array} \end{array}$$

Based on the properties of the rotation matrix and the above expression, it can be seen that ${\hat{\Phi }}^{-1}$ and $\Phi^{-1}$ are similarity matrices. Thus,
(36)$$\begin{array}{c} \begin{array}{c} \mathrm{tr}\left({\hat{\Phi }}^{-1}\right)=\mathrm{tr}\left(\Phi^{-1}\right). \end{array} \end{array}$$

Thus, we can conclude that 3D rotations do not affect the $\mathrm{tr}\left(\Phi^{-1}\right)$ calculated from the AOA-based FIM. In the next section, we will derive the optimal sensor placement with Gaussian priors using the invariance of the trace of FIM to 3D rotations.

## 4. Optimal Sensor Placement with Gaussian Priors

In this section, we investigate the optimal sensor placement with Gaussian priors. First, the FIM for 3D AOA localization with Gaussian priors is derived, and the solution of minimizing the trace of CRLB is developed. Moreover, Section 3 provided a solution for diagonalizing $P_{0}$ with proper 3D rotation. The invariance property for 3D rotation of the AOA-based $\mathrm{tr}\left(\Phi^{-1}\right)$ is used to diagonalize the non-diagonal covariance. Therefore, we suppose that the coordinate system is rotated such that the covariance matrix is diagonal $P_{0}\;=\;\mathrm{diag}\left(\left[a,b,c\right]\right)$.

Based on (Equation 11), the FIM for the 3D AOA target localization problem is
(37)$$\begin{array}{c} \begin{array}{cc} \Phi & =P_{0}^{-1}+{J_{2}}^{T}\Sigma^{-1}J_{2}=P_{0}^{-1}+\sum_{k=1}^{N}\frac{1}{r_{k}^{2}\sigma_{\theta_{k}}^{2}\cos^{2}\phi_{k}}u_{k}u_{k}^{T}+\sum_{k=1}^{N}\frac{1}{r_{k}^{2}\sigma_{\phi_{k}}^{2}}v_{k}v_{k}^{T}, \end{array} \end{array}$$
where $u_{k}$ and $v_{k}$ are unit vectors orthogonal to the 2D azimuth vector and 3D range vector, respectively,
(38)$$\begin{array}{c} \begin{array}{c} u_{k}=\left[\begin{array}{c} -\sin\theta_{k}\\ \cos\theta_{k}\\ 0 \end{array}\right],\;v_{k}=\left[\begin{array}{c} -\sin\phi_{k}\cos\theta_{k}\\ -\sin\phi_{k}\sin\theta_{k}\\ \cos\phi_{k} \end{array}\right]. \end{array} \end{array}$$

Following (Equation 19), we aim to determine optimal sensor locations, and the optimality criterion is to minimize the trace of CRLB, which is also known as the optimality criterion [35]. This section first investigates the optimal palcement of one sensor and then expands to multiple sensors.

### 4.1. Optimal Sensor Placement for One Sensor

Let us discuss the optimal placement for one sensor with Gaussian priors. Substitute (Equation 38) into (Equation 37) and then use (Equation 19). Then we can see that the trace of CRLB satisfies
(39)$$\begin{array}{c} \begin{array}{cc} \mathrm{tr}\left(\mathrm{CRLB}\right)\; & =\mathrm{tr}\left(\Phi^{-1}\right)\geq {\left(a^{-1}+\frac{1}{r^{2}}\left(\frac{\sin^{2}\theta }{\sigma_{\theta }^{2}\cos^{2}\phi }+\frac{1}{\sigma_{\phi }^{2}}\sin^{2}\phi \cos^{2}\theta \right)\right)}^{-1}\\ \; & +{\left(b^{-1}+\frac{1}{r^{2}}\left(\frac{\cos^{2}\theta }{\sigma_{\theta }^{2}\cos^{2}\phi }+\frac{1}{\sigma_{\phi }^{2}}\sin^{2}\phi \sin^{2}\theta \right)\right)}^{-1}+{\left(c^{-1}+\frac{\cos^{2}\phi }{\sigma_{\phi }^{2}r^{2}}\right)}^{-1}, \end{array} \end{array}$$
with equality if
(40)$$\begin{array}{c} \begin{array}{cc} \; & -\frac{\sin2\theta }{\sigma_{\theta }^{2}\cos^{2}\phi }+\frac{1}{\sigma_{\phi }^{2}}\sin^{2}\phi \sin2\theta =0,\\ \; & \frac{1}{\sigma_{\phi }^{2}}\sin2\phi \cos\theta =0,\;\frac{1}{\sigma_{\phi }^{2}}\sin2\phi \sin\theta =0. \end{array} \end{array}$$

To satisfy the above expression, we compute the azimuth and elevation angle as follows:(41)$$\begin{array}{c} \begin{array}{c} \left\{\theta ,\phi \right\}\in \left\{\left\{\pm \pi /2\;,0\right\},\left\{\pm \pi /2\;,\pm \pi /2\;\right\},\left\{0,0\right\},\left\{0,\pm \pi /2\;\right\}\right\}. \end{array} \end{array}$$

Substituting the optimal angle $\left\{\theta ,\phi \right\}$ into (Equation 39), we can obtain different configurations, as listed in Table 1. We set R1=a, R2=b, R3=c, $R4=r^{2}\sigma_{\theta }^{2}$, and $R5=r^{2}\sigma_{\phi }^{2}$, then adopt the resistor network model to find the minimum tr(CRLB), which depends on the prior covariance matrices, the angle noise variances $\sigma_{\theta }$ and $\sigma_{\phi }$, and the sensor-target ranges *r*. The resistor network model for optimal sensor placement with different configurations is shown in Figure 3.

Furthermore, the resistor networks can help determine the optimal geometry rapidly using the analysis of different configurations, and the value of a,b,c with the prior covariance matrix $P_{0}$ mainly decides the optimal placement when $r^{2}\sigma_{\phi }^{2}$ and $r^{2}\sigma_{\theta }^{2}$ are fixed by using the parallel resistor equation. The explanation of configurations in Table 1:Configuration 1: The values of resistors R1 and R2 can be reduced owing to the parallel resistors R4 and R5. Thus, the angle is suited for a>c>b and c>a>b.Configuration 2: The value of resistor R1 is eliminated, so the angle is suited for a>b>c.Configuration 3: The value of resistor R2, R3 can be reduced owing to the parallel resistors R4 and R5. Thus, the angle is suited for b>c>a, c>b>a.Configuration 4: The value of resistor R2 is eliminated, so the angle is suited for b>a>c.

In conclusion, when the maximum value is *a*, the optimal angle of $\left\{\theta ,\phi \right\}$ is $\left\{\pm \pi /2,0\right\}$, $\left\{\pm \pi /2,\pm \pi /2\right\}$, and the line of sight (LOS) $\left\{{\left[0,1,0\right]}^{T},{\left[0,0,1\right]}^{T}\right\}$ is orthogonal to the largest eigenvector of $P_{0}$. A similar conclusion can be derived when the maximum value is *b* or *c*, which has the same results as [26]. Moreover, the non-diagonal covariance placement can easily be attained using the above analytical finding. This method is much simpler than the sensor update method in [26].

### 4.2. Optimal Sensor Placement for N=2

In this subsection, we consider the case of two sensors and use the resistor network model to determine the optimal sensor placement. Substituting N=2 into (Equation 37), the trace of inverse of FIM is written as
(42)$$\begin{array}{c} \begin{array}{cc} \; & \mathrm{tr}\left(\mathrm{CRLB}\right)=\mathrm{tr}\left(\Phi^{-1}\right)\geq {\left(a^{-1}+\sum_{k=1}^{2}\frac{1}{r_{k}^{2}}\left(\frac{\sin^{2}\theta_{k}}{\sigma_{\theta_{k}}^{2}\cos^{2}\phi_{k}}+\frac{1}{\sigma_{\phi_{k}}^{2}}\sin^{2}\phi_{k}\cos^{2}\theta_{k}\right)\right)}^{-1}\\ \; & +{\left(b^{-1}+\sum_{k=1}^{2}\frac{1}{r_{k}^{2}}\left(\frac{\cos^{2}\theta_{k}}{\sigma_{\theta_{k}}^{2}\cos^{2}\phi_{k}}+\frac{1}{\sigma_{\phi_{k}}^{2}}\sin^{2}\phi_{k}\sin^{2}\theta_{k}\right)\right)}^{-1}+{\left(c^{-1}+\sum_{k=1}^{2}\left(\frac{\cos^{2}\phi_{k}}{\sigma_{\phi_{k}}^{2}r_{k}^{2}}\right)\right)}^{-1}, \end{array} \end{array}$$
with equality if
(43)$$\begin{array}{c} \begin{array}{cc} \; & \sum_{k=1}^{2}\frac{1}{r_{k}^{2}}\left(\frac{1}{\sigma_{\phi_{k}}^{2}}\sin^{2}\phi_{k}\sin2\theta_{k}-\frac{\sin2\theta_{k}}{\sigma_{\theta_{k}}^{2}\cos^{2}\phi_{k}}\right)=0,\\ \; & \sum_{k=1}^{2}\frac{1}{r_{k}^{2}\sigma_{\phi_{k}}^{2}}\sin2\phi_{k}\cos\theta_{k}=0,\;\sum_{k=1}^{2}\frac{1}{r_{k}^{2}\sigma_{\phi_{k}}^{2}}\sin2\phi_{k}\sin\theta_{k}=0. \end{array} \end{array}$$

For azimuth angles, the two-sensor optimal placement in the 2D plane that minimizes the tr(CRLB) is given by $\left|\theta_{1}-\theta_{2}\right|=\pi /2$, regardless of noise variance and sensor ranges [23]. Since we set $\left\{\theta_{1},\theta_{2}\right\}=\left\{0,\pm \pi /2\right\}$, and the above equations can be satisfied when
(44)$$\begin{array}{c} \begin{array}{c} \left\{\phi_{1},\phi_{2}\right\}\in \left\{\left\{0,0\right\},\left\{0,\pm \pi /2\right\},\left\{\pm \pi /2,0\right\},\left\{\pm \pi /2,\pm \pi /2\right\}\right\}. \end{array} \end{array}$$

By substituting (Equation 44) into (Equation 42), we can obtain the tr(CRLB) for $\left\{\theta_{1},\theta_{2}\right\}=\left\{0,\pm \pi /2\right\}$ with different elevation angles that listed in Table 2. Besides, we set R1=a, R2=b, R3=c, $R4=r_{1}^{2}\sigma_{\theta 1}^{2}$, $R5=r_{2}^{2}\sigma_{\theta 2}^{2}$, $R6=r_{1}^{2}\sigma_{\phi 1}^{2}$, and $R7=r_{2}^{2}\sigma_{\phi 2}^{2}$. The minimum trace of CRLB depends on the prior covariance matrix, the angle noise variances, and the sensor-target ranges. The resistor network model for optimal sensor placement with the different configurations is shown in Figure 4.

### 4.3. Optimal Sensor Placement for N≥3

In this section, we consider the optimal placement of *N* sensors in 3D space with different angle noises and distances. The trace of inverse of FIM is written as
(45)$$\begin{array}{c} \begin{array}{cc} \; & \mathrm{tr}\left(\mathrm{CRLB}\right)=\mathrm{tr}\left(\Phi^{-1}\right)\geq {\left(a^{-1}+\sum_{k=1}^{N}\frac{1}{r_{k}^{2}}\left(\frac{\sin^{2}\theta_{k}}{\sigma_{\theta_{k}}^{2}\cos^{2}\phi_{k}}+\frac{1}{\sigma_{\phi_{k}}^{2}}\sin^{2}\phi_{k}\cos^{2}\theta_{k}\right)\right)}^{-1}\\ \; & +{\left(b^{-1}+\sum_{k=1}^{N}\frac{1}{r_{k}^{2}}\left(\frac{\cos^{2}\theta_{k}}{\sigma_{\theta_{k}}^{2}\cos^{2}\phi_{k}}+\frac{1}{\sigma_{\phi_{k}}^{2}}\sin^{2}\phi_{k}\sin^{2}\theta_{k}\right)\right)}^{-1}+{\left(c^{-1}+\sum_{k=1}^{N}\left(\frac{\cos^{2}\phi_{k}}{\sigma_{\phi_{k}}^{2}r_{k}^{2}}\right)\right)}^{-1}, \end{array} \end{array}$$
subject to
(46)$$\begin{array}{c} \begin{array}{cc} \; & \sum_{k=1}^{N}\frac{1}{r_{k}^{2}}\left(\frac{1}{\sigma_{\phi_{k}}^{2}}\sin^{2}\phi_{k}\sin2\theta_{k}-\frac{\sin2\theta_{k}}{\sigma_{\theta_{k}}^{2}\cos^{2}\phi_{k}}\right)=0,\\ \; & \sum_{k=1}^{N}\frac{1}{r_{k}^{2}\sigma_{\phi_{k}}^{2}}\sin2\phi_{k}\cos\theta_{k}=0,\;\sum_{k=1}^{N}\frac{1}{r_{k}^{2}\sigma_{\phi_{k}}^{2}}\sin2\phi_{k}\sin\theta_{k}=0. \end{array} \end{array}$$

To diagonalize FIM, the azimuth and elevation angle can be shown to obey the following equality [21]:(47)$$\begin{array}{c} \begin{array}{c} \sin2\theta_{k}=0,k=1,\ldots ,N,\;\sin2\phi_{k}=0,k=1,\ldots ,N. \end{array} \end{array}$$

Define the subset of C as the optimal azimuth angles, which is given by
(48)$$\begin{array}{c} \begin{array}{c} C=\left\{\left\{\theta_{1},\theta_{2},\ldots ,\theta_{N}\right\}\left|\theta_{k}\in \left\{0,\pm \pi /2\right\},k=1,\ldots ,N\right.\right\}, \end{array} \end{array}$$

The elevation angles satisfy (Equation 45) form a set defined as
(49)$$\begin{array}{c} \begin{array}{c} Z=\left\{\left\{\phi_{1},\phi_{2},\ldots \phi_{N}\right\}\left|\phi_{k}\in \left\{0,\pm \pi /2\right\},k=1,\ldots ,N\right.\right\}. \end{array} \end{array}$$

Thus, we can get the minimum trace of CRLB with the angle combination of C and Z.
(50)$$\begin{array}{c} \begin{array}{cc} \; & \mathrm{tr}\left(\Phi_{\mathrm{opt}}^{-1}\left(\theta_{1},\ldots ,\theta_{N},\phi_{1},\ldots ,\phi_{N}\right)\right)={\left(a^{-1}+\sum_{k=1}^{N}\frac{1}{r_{k}^{2}}\left(\frac{\sin^{2}\theta_{k}}{\sigma_{\theta_{k}}^{2}\cos^{2}\phi_{k}}+\frac{1}{\sigma_{\phi_{k}}^{2}}\sin^{2}\phi_{k}\cos^{2}\theta_{k}\right)\right)}^{-1}\\ \; & +{\left(b^{-1}+\sum_{k=1}^{N}\frac{1}{r_{k}^{2}}\left(\frac{\cos^{2}\theta_{k}}{\sigma_{\theta_{k}}^{2}\cos^{2}\phi_{k}}+\frac{1}{\sigma_{\phi_{k}}^{2}}\sin^{2}\phi_{k}\sin^{2}\theta_{k}\right)\right)}^{-1}+{\left(c^{-1}+\sum_{k=1}^{N}\left(\frac{\cos^{2}\phi_{k}}{\sigma_{\phi_{k}}^{2}r_{k}^{2}}\right)\right)}^{-1}. \end{array} \end{array}$$

Therefore, (Equation 48) and (Equation 49) can be used to determine the optimal sensor placement N≥3.

Based on the analysis above, we can get the optimal azimuth and elevation angles subset. This conclusion is consistent with the literature [21]. In addition, it can be seen that the parameters of $P_{0}$ also affect the sensor placement with the analysis of the resistor network models. Therefore, the minimum trace of CRLB depends on the angle noise variances, the sensor-target distance, and the value of $P_{0}$.

## 5. Simulation Studies

### 5.1. Gradient Descent Alogorithm Simulations

In this subsection, we adopt a gradient descent algorithm to verify the optimal sensor placement conditions derived in the above section. Assume that the distribution of target is given, and $s_{0}={\left(0,0,0\right)}^{T}$. The minimum distances between the target and sensors are represented by $d_{k}$. A group of mobile sensors is moving to minimize the trace of CRLB in 3D space [21]. This exact gradient descent simulation was run 10,000 steps.

Example 1: For optimal sensor placement with one sensor

Case A: We used these simulation parameters: $P_{0}=\left[\begin{array}{ccc} 500 & 0 & 0\\ 0 & 200 & 0\\ 0 & 0 & 100 \end{array}\right]$, d=150 m, $\sigma_{\theta }^{2}=\sigma_{\phi }^{2}=1^{\circ }$, and the initial sensor location was ${\left[\begin{array}{ccc} 200 & -100 & -100\sqrt{2} \end{array}\right]}^{T}$. The sensor trajectory is shown in Figure 5a, and the final angles were $\theta =-91.34^{\circ }$ and $\phi =-89.53^{\circ }$, which matches Configuration 2 (a>b>c) in Table 1, and the LOS was orthogonal to the largest eigenvector of $P_{0}$.

Case B: The simulation parameters were as follows: $P_{0}=\left[\begin{array}{ccc} 100 & 0 & 0\\ 0 & 200 & 0\\ 0 & 0 & 500 \end{array}\right]$, d=200 m, $\sigma_{\theta }^{2}=1^{\circ }$, $\sigma_{\phi }^{2}=2^{\circ }$, and the initial sensor location was ${\left[\begin{array}{ccc} 100 & 200 & 100\sqrt{2} \end{array}\right]}^{T}$. The sensor trajectory is shown in Figure 5b and the final angles were $\theta =-0.03^{\circ }$ and $\phi =-0.02^{\circ }$, which matches Configuration 3 (c>b>a) in Table 1, and the LOS was orthogonal to the largest eigenvector of $P_{0}$. Moreover, although the initial sensor location and *d* were different in Cases A and B, it is shown that the final optimal sensor placement also matches the analysis results in Figure 5a,b. The simulation results also can prove the proposed method without any restriction on the sensor-target range and initial sensor locations.

Case C: We used the parameters of Case B except $P_{0}=\left[\begin{array}{ccc} 100 & 50 & 20\\ 50 & 200 & 30\\ 20 & 30 & 500 \end{array}\right]$. The rotation angles were computed using (Equation 20) and (Equation 21), i.e., $\alpha =7.10^{\circ }$, $\beta =358.88^{\circ }$, γ=22.26, and $P_{0}$ can be rewritten as $P_{0}^{r}=\left[\begin{array}{ccc} 79.17 & 0 & 0\\ 0 & 216.31 & 0\\ 0 & 0 & 504.52 \end{array}\right]$. The rest of simulation parameters can be obtained from (Equation 22), and the tr(CRLB) was computed using (Equation 33). The sensor trajectory is shown in Figure 5c, and the final angles were $\theta =-0.03^{\circ }$ and $\phi =-0.01^{\circ }$, which matches Configuration 3 (c>b>a) in Table 1. The LOS was orthogonal to the largest eigenvector of $P_{0}$.

Case D: We used the parameters of Case B except $P_{0}=\left[\begin{array}{ccc} 300 & 10 & 20\\ 10 & 500 & 15\\ 20 & 15 & 100 \end{array}\right]$, and $P_{0}^{r}=\left[\begin{array}{ccc} 301.46 & 0 & 0\\ 0 & 501.13 & 0\\ 0 & 0 & 97.41 \end{array}\right]$ after the 3D rotation. As in Case C, the final angles were $\theta =-0.16^{\circ }$ and $\phi =-89.67^{\circ }$, which matches Configuration 4 (b>a>c) in Table 1; and the LOS was orthogonal to the largest eigenvector of $P_{0}$, and the sensor trajectory is shown in Figure 5d.

More specifically, the prior covariance matrices $P_{0}$ in Cases A and B were diagonal covariance matrices $P_{0}$. We could quickly obtain the optimal placement through the gradient simulation, and the results of Figure 5a,b match the findings in Section 4.1. Besides, the prior covariance matrices $P_{0}$ in Cases C and D were non-diagonal covariance matrices, and the invariance property for 3D rotation of the AOA-based trace of CRLB was used to diagonalize the non-diagonal covariance. Then, we obtained the optimal placement using the gradient simulation, and the results of Figure 5c,d also match the findings in Section 4.1.

Example 2: Optimal sensor placement for two and three sensors:

Case A: The simulation parameters were as follows: $P_{0}=\left[\begin{array}{ccc} 200 & 0 & 0\\ 0 & 600 & 0\\ 0 & 0 & 900 \end{array}\right]$, $d_{1}=d_{2}=200$ m, $\sigma_{\theta_{1}}^{2}=\sigma_{\theta_{2}}^{2}=0.5^{\circ },\sigma_{\phi_{1}}^{2}=\sigma_{\phi_{2}}^{2}=1^{\circ }$, and the initial sensor locations were ${\left[\begin{array}{ccc} 200 & -100 & -100\sqrt{2} \end{array}\right]}^{T}$, ${\left[\begin{array}{ccc} 100 & -100 & 200 \end{array}\right]}^{T}$. The sensors’ trajectories are shown in Figure 6a, and the final angles were $\theta_{1}=-37.24^{\circ }$, $\theta_{2}=-130.29^{\circ }$, $\phi_{1}=-0.03^{\circ }$, and $\phi_{2}=88.91^{\circ }$, which matches Configuration 2 in Table 2.

Case B: We used the parameters of Case A except $P_{0}=\left[\begin{array}{ccc} 200 & 20 & 15\\ 20 & 600 & 50\\ 15 & 50 & 900 \end{array}\right]$, and $P_{0}^{r}=\left[\begin{array}{ccc} 198.52 & 0 & 0\\ 0 & 596.23 & 0\\ 0 & 0 & 905.25 \end{array}\right]$ after rotation. The sensors’ trajectories are shown in Figure 6b, and the final angles were $\theta_{1}=-27.95^{\circ }$, $\theta_{2}=-116.51^{\circ }$, $\phi_{1}=-0.06^{\circ }$, and $\phi_{2}=88.65^{\circ }$, which also matches Configuration 2 in Table 2.

In Cases A and B, we adopted the same parameters except for the covariance matrix $P_{0}$. Similarly, the non-diagonal covariance matrix in Case B was diagonalized by the 3D rotation. It is shown that the sensor trajectories and the final optimal sensor-target geometries were almost identical in Figure 6a,b, which satisfies the results of Section 4.2.

Case C: For three sensors, we used the simulation parameters as follows: $P_{0}=\left[\begin{array}{ccc} 300 & 0 & 0\\ 0 & 800 & 0\\ 0 & 0 & 900 \end{array}\right]$, $d_{1}=d_{2}=d_{3}=200$ m, $\sigma_{\theta_{1}}^{2}=\sigma_{\theta_{2}}^{2}=\sigma_{\theta_{3}}^{2}=0.5^{\circ },\sigma_{\phi_{1}}^{2}=\sigma_{\phi_{2}}^{2}=\sigma_{\phi_{3}}^{2}=0.5^{\circ }$, and the initial sensor locations were ${\left[\begin{array}{ccc} -100\sqrt{2} & 100 & -200 \end{array}\right]}^{T}$, ${\left[\begin{array}{ccc} 100 & -100\sqrt{2} & 0 \end{array}\right]}^{T}$, ${\left[\begin{array}{ccc} -100\sqrt{2} & 100 & 200 \end{array}\right]}^{T}$. The sensors’ trajectories are shown in Figure 6c and the final angles were $\theta_{1}=118.51^{\circ }$, $\theta_{2}=-61.49^{\circ }$, $\theta_{3}=-155.64^{\circ }$, $\phi_{1}=-0.01^{\circ }$, $\phi_{2}=0.01^{\circ }$, and $\phi_{3}=88.91^{\circ }$.

Case D: We used the parameters of Case C except $P_{0}=\left[\begin{array}{ccc} 300 & 15 & 20\\ 15 & 800 & 30\\ 20 & 30 & 900 \end{array}\right]$, and $P_{0}^{r}=\left[\begin{array}{ccc} 299.18 & 0 & 0\\ 0 & 795.77 & 0\\ 0 & 0 & 905.05 \end{array}\right]$ after rotation. The sensors’ trajectories are shown in Figure 6d, and the final angles were $\theta_{1}=120.51^{\circ }$, $\theta_{2}=-59.49^{\circ }$, $\theta_{3}=-146.41^{\circ }$, $\phi_{1}=0.05^{\circ }$, $\phi_{2}=0.01^{\circ }$, and $\phi_{3}=88.33^{\circ }$.

Similarly, we used the same parameters except for the covariance matrix $P_{0}$ in Cases C and D. The non-diagonal covariance matrix in Case D was diagonalized by the 3D rotation. It is shown that the sensor trajectories and the final optimal sensor-target geometries were almost identical in Figure 6c,d, which also satisfies the results of Section 4.3.

For Cases A and B in Example 2, the $\mathrm{tr}\left(\Phi_{\mathrm{opt}}^{-1}\right)$ computed by the gradient descent algorithm were approximately the same; besides, we could obtain the theoretical minimum trace of CRLB using (Equation 42) with the optimal sensor placement. The $\mathrm{tr}(\Phi^{-1})$ from Case A and $\mathrm{tr}\left({\hat{\Phi }}^{-1}\right)$ from Case B were equal, which is in agreement with the analytical result of (Equation 36). Table 3 lists the $\mathrm{tr}\left(\Phi_{\mathrm{opt}}^{-1}\right)$, $\mathrm{tr}(\Phi^{-1})$ and $\mathrm{tr}\left({\hat{\Phi }}^{-1}\right)$ for different cases of Example 2. It is clear that the same conclusion was obtained for N=3 in Example 2 for Cases C and D. Furthermore, the $\mathrm{tr}\left(\Phi_{\mathrm{opt}}^{-1}\right)$ is close to the theoretical minimum trace; i.e., $\mathrm{tr}(\Phi^{-1})$ and $\mathrm{tr}\left({\hat{\Phi }}^{-1}\right)$.

### 5.2. The Comparison Results

This subsection demonstrates the optimal sensor placement with the maximum a posteriori (MAP) estimation simulations, and the MAP is deduced in Appendix A. In the example, the method in [21] and the method in [22] using the D-optimality criterion are compared with the proposed method. In this paper, we use “the method in [21]” and “the method in [22]” to denote the optimal placement methods in [21,22], respectively. The parameters were as follows: $s_{0}={\left(0,0,0\right)}^{T}$, $P_{0}=\left[\begin{array}{ccc} 100 & 0 & 0\\ 0 & 200 & 0\\ 0 & 0 & 800 \end{array}\right]$, and the initial sensor locations were ${\left[\begin{array}{ccc} -200 & -100 & -100\sqrt{2} \end{array}\right]}^{T}$, ${\left[\begin{array}{ccc} 100 & -100\sqrt{2} & -200 \end{array}\right]}^{T}$, ${\left[\begin{array}{ccc} -100\sqrt{2} & 100 & 200 \end{array}\right]}^{T}$, ${\left[\begin{array}{ccc} 100\sqrt{2} & 200 & -100\sqrt{2} \end{array}\right]}^{T}$. We added different noise levels and show the theoretical minimum trace of CRLBs and MSEs; i.e., $\sigma_{\theta_{1}}^{2}=\sigma_{\theta_{2}}^{2}=\sigma_{\theta_{3}}^{2}=\sigma_{\theta_{4}}^{2}=0.5^{\circ }$, and $\sigma_{\phi_{1}}^{2}=\sigma_{\phi_{2}}^{2}=\sigma_{\phi_{3}}^{2}=\sigma_{\phi_{4}}^{2}$, the value of $\sigma_{\phi }^{2}$ from $0.2^{\circ }$ to $1.8^{\circ }$.

The theoretical trace of CRLBs and MSEs of different sensor placements are shown in Figure 7. The MSEs of MAP were estimated using 10,000 Monte Carlo simulations. The MAP estimator was implemented using the Gauss–Newton method and initialized to the prior mean target location $s_{0}$. The results showed that the optimal sensor placement can always provide better MSEs than the other existing methods.

Next, we compare the localization accuracies of different methods. We fixed N=3, $\sigma_{\theta }^{2}=0.5^{\circ }$ and increased the value of $\sigma_{\phi }^{2}$ from $0.1^{\circ }$ to $1^{\circ }$. The settings of others parameters were the same as in Case C of Example 2. The optimal angles in [21] are $\theta_{1}=0^{\circ }$, $\theta_{2}=90^{\circ }$, $\theta_{3}=-90^{\circ }$, $\phi_{1}=0^{\circ }$, $\phi_{2}=0^{\circ }$, and $\phi_{3}=0^{\circ }$. The correspondingly optimal angles were adopted in Case C of Example 2 as $\theta_{1}=118^{\circ }$, $\theta_{2}=-62^{\circ }$, $\theta_{3}=-152^{\circ }$, $\phi_{1}=0^{\circ }$, $\phi_{2}=0^{\circ }$, and $\phi_{3}=90^{\circ }$. Figure 8 shows the comparison of tr(CRLB)s computed by the method in [21], the method in [22], and the final sensor locations in Case C of Example 2.

From Figure 8, it can be seen that the proposed method in this paper had better estimation performance than the existing methods, even if both the proposed method and the method in [21] contained optimal azimuth and elevation angles subsets. This result also can confirm the analytical optimal sensor placement in Section 4.

Finally, we compare the method in [21,22] in terms of estimation performance for different sensor numbers. The sensors started from different original locations, and we set $P_{0}=\left[\begin{array}{ccc} 200 & 0 & 0\\ 0 & 500 & 0\\ 0 & 0 & 700 \end{array}\right]$, d=200 m, $\sigma_{\theta }^{2}=\sigma_{\phi }^{2}=1^{\circ }$. Table 4 lists the MSEs and bias norms when the number of sensors is N=3,4,5,6. Due to the effect of the prior covariance matrix, the performance of the existing methods was worse than that of the proposed method. The MSEs of our proposed method were much smaller than those of the existing methods with the different sensor numbers. From Figure 8 and Table 4, we conclude that the proposed method can achieve the optimal estimation performance.

## 6. Conclusions

In this paper, an optimal sensor placement method for an uncertain target with Gaussian priors was presented. Our analysis was conducted based on minimizing the trace of the inverse FIM. The invariance property for the 3D rotation of the AOA-based FIM was provided, which can be used to diagonalize the non-diagonal covariance matrix. An optimal sensor placement analysis for the 3D space with the diagonal covariance matrix of the target was presented, and a resistor network was used to represent the optimal sensor placement strategy. It was demonstrated that the optimal localization placements have a similar geometric configuration, regardless of the diagonality of the covariance matrix. Finally, the analytical results were verified via a series of numerical simulations. The analytical and numerical findings coincide with the simulation results.

For future work, we will consider a case with multiple uncertain targets with different Gaussian priors, which changes the optimization problem to a convex combination of FIMs. In addition, the optimal trajectories also can be developed for the uncertain moving target with Gaussian priors. 

## Figures and Tables

**Figure 1 entropy-23-01379-f001:**
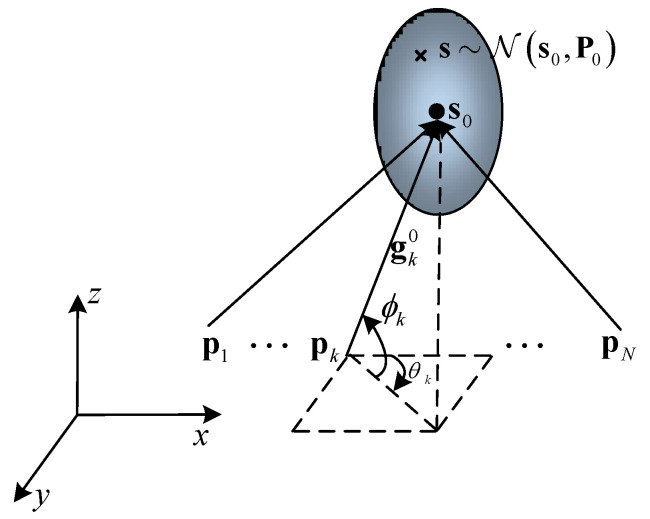
3D AOA localization sensor placement with Gaussian priors.

**Figure 2 entropy-23-01379-f002:**
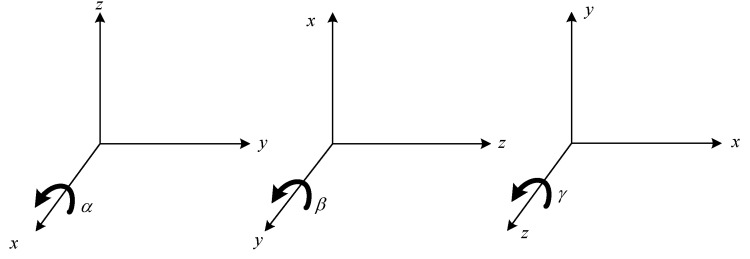
The rotation angles α, β, γ around the *x*, *y*, and *z* axes.

**Figure 3 entropy-23-01379-f003:**
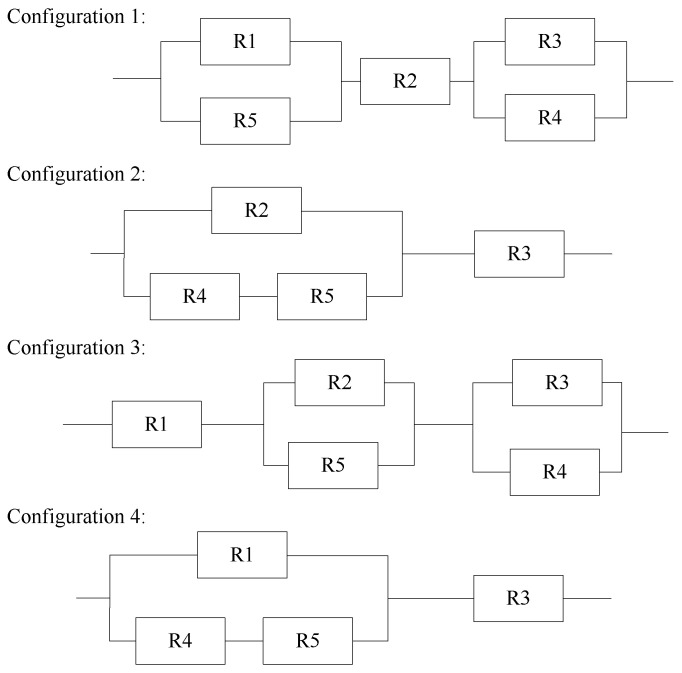
Resistor network model for optimal sensor placement for one sensor.

**Figure 4 entropy-23-01379-f004:**
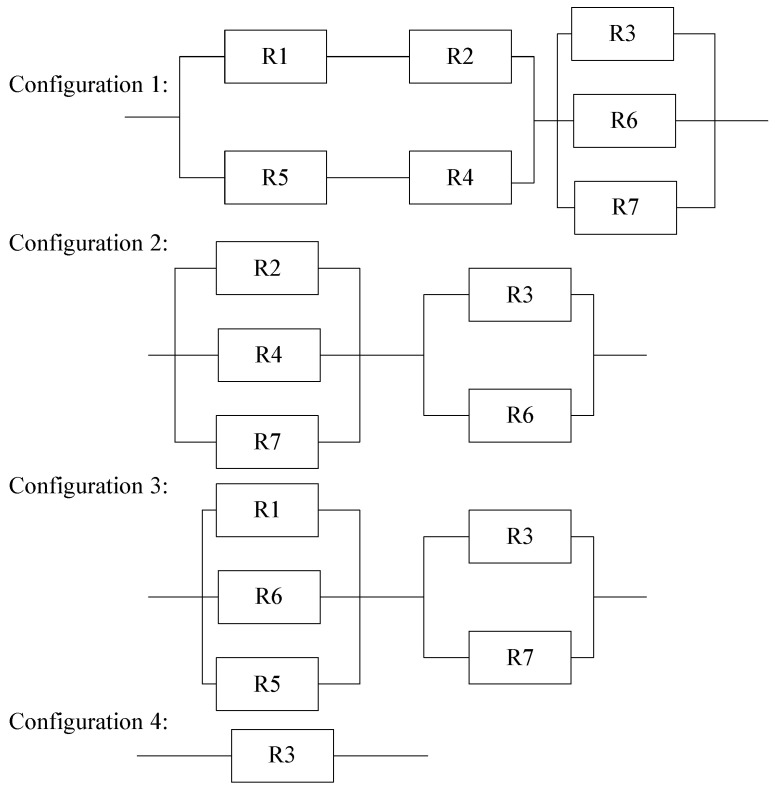
Resistor network model for optimal sensor placement for N=2.

**Figure 5 entropy-23-01379-f005:**
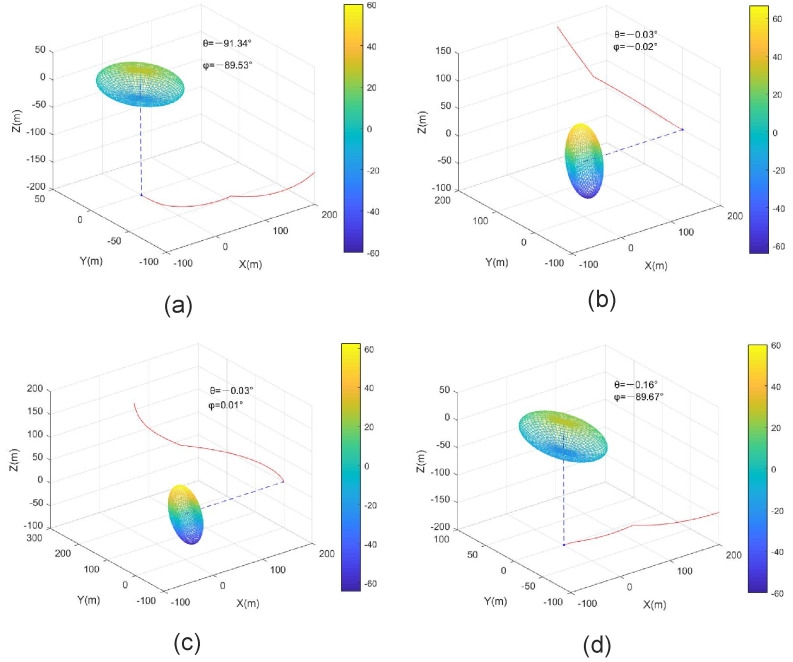
Optimal sensor placement for one sensor. (**a**) $P_{0}$ is a diagonal matrix with a>b>c, (**b**) $P_{0}$ is a diagonal matrix with c>b>a, (**c**) $P_{0}$ is a non-diagonal matrix with c>b>a, (**d**) $P_{0}$ is a non-diagonal matrix with b>a>c.

**Figure 6 entropy-23-01379-f006:**
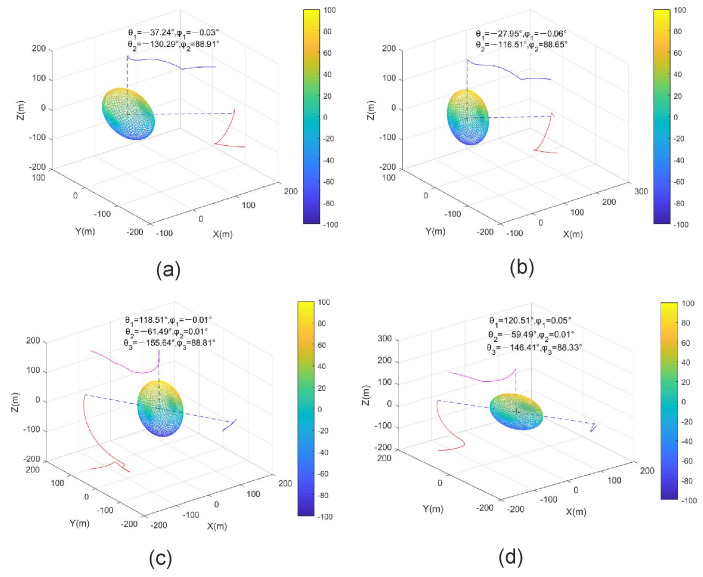
Optimal sensor placement with two and three sensors. (**a**) $P_{0}$ is a diagonal matrix with N=2, (**b**) $P_{0}$ is a non-diagonal matrix with N=2, (**c**) $P_{0}$ is a diagonal matrix with N=3, (**d**) $P_{0}$ is a non-diagonal matrix with N=3.

**Figure 7 entropy-23-01379-f007:**
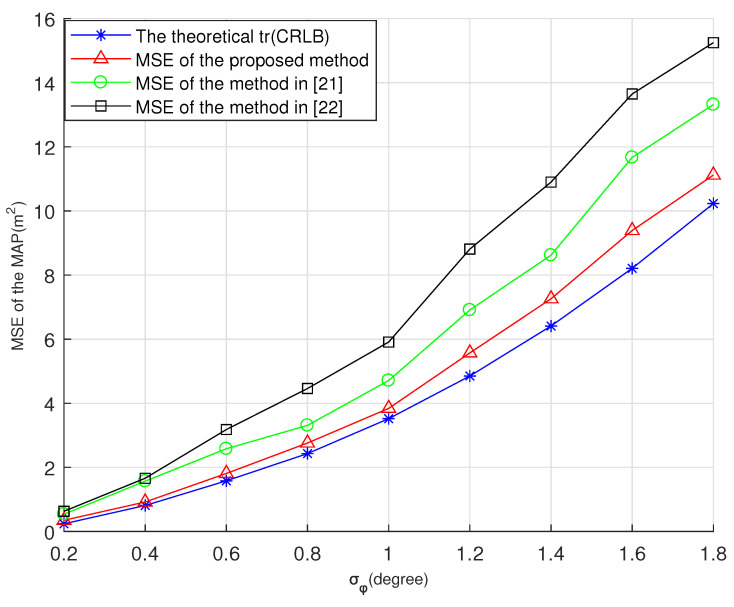
Estimation comparison with $\sigma_{\theta }^{2}=0.5^{\circ }$ and $\sigma_{\phi }^{2}=0.2^{\circ }$ to $1.8^{\circ }$.

**Figure 8 entropy-23-01379-f008:**
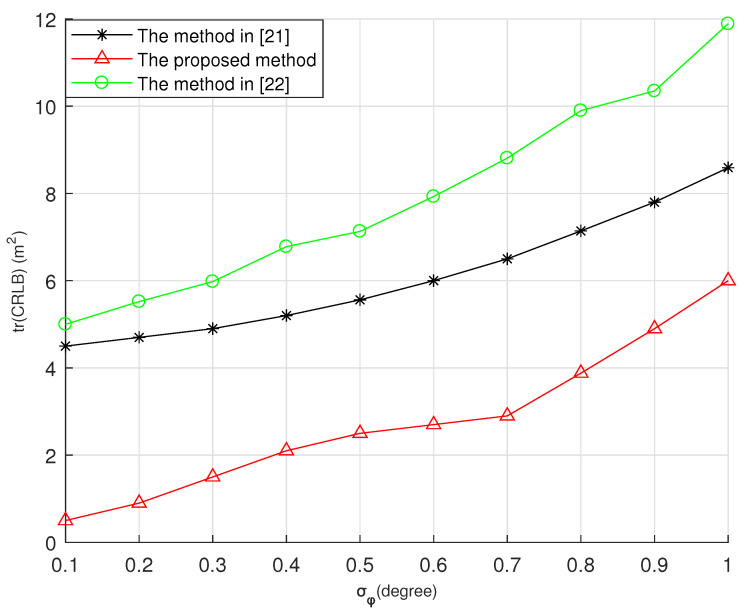
The comparison results with $\sigma_{\theta }^{2}=1^{\circ }$ and $\sigma_{\phi }^{2}=0.1^{\circ }$ to $1^{\circ }$.

**Table 1 entropy-23-01379-t001:** Trace of CRLB with different optimal angles and configurations.

Configuration	θ	ϕ	tr(CRLB)
1	±π/2	0	${\left(a^{-1}+\frac{1}{r^{2}\sigma_{\theta }^{2}}\right)}^{-1}+b+{\left(c^{-1}+\frac{1}{r^{2}\sigma_{\phi }^{2}}\right)}^{-1}$
2	±π/2	±π/2	${\left(b^{-1}+\frac{1}{r^{2}\sigma_{\theta }^{2}}+\frac{1}{r^{2}\sigma_{\phi }^{2}}\right)}^{-1}+c$
3	0	0	$a+{\left(b^{-1}+\frac{1}{r^{2}\sigma_{\theta }^{2}}\right)}^{-1}+{\left(c^{-1}+\frac{1}{r^{2}\sigma_{\phi }^{2}}\right)}^{-1}$
4	0	±π/2	${\left(a^{-1}+\frac{1}{r^{2}\sigma_{\theta }^{2}}+\frac{1}{r^{2}\sigma_{\phi }^{2}}\right)}^{-1}+c$

**Table 2 entropy-23-01379-t002:** Trace of CRLB for $\left\{\theta_{1},\theta_{2}\right\}=\left\{0,\pm \pi /2\right\}$ and different elevation-angles.

Configuration	$\phi_{1}$	$\phi_{2}$	tr(CRLB)
1	0	0	${\left(a^{-1}+\frac{1}{r_{2}^{2}\sigma_{\theta_{2}}^{2}}\right)}^{-1}+{\left(b^{-1}+\frac{1}{r_{1}^{2}\sigma_{\theta_{1}}^{2}}\right)}^{-1}+{\left(c^{-1}+\frac{1}{r_{1}^{2}\sigma_{\phi_{1}}^{2}}+\frac{1}{r_{2}^{2}\sigma_{\phi_{2}}^{2}}\right)}^{-1}$
2	0	±π/2	${\left(b^{-1}+\frac{1}{r_{1}^{2}\sigma_{\theta_{1}}^{2}}+\frac{1}{r_{2}^{2}\sigma_{\phi_{2}}^{2}}\right)}^{-1}+{\left(c^{-1}+\frac{1}{r_{1}^{2}\sigma_{\phi_{1}}^{2}}\right)}^{-1}$
3	±π/2	0	${\left(a^{-1}+\frac{1}{r_{1}^{2}\sigma_{\phi_{1}}^{2}}+\frac{1}{r_{2}^{2}\sigma_{\theta_{2}}^{2}}\right)}^{-1}+{\left(c^{-1}+\frac{1}{r_{2}^{2}\sigma_{\phi_{2}}^{2}}\right)}^{-1}$
4	±π/2	±π/2	*c*

**Table 3 entropy-23-01379-t003:** Trace of CRLB for Example 2.

Example 2	$\mathrm{tr}\left(\Phi_{\mathrm{opt}}^{-1}\right)$ $\left(m^{2}\right)$	$\mathrm{tr}(\Phi^{-1})$ $\left(m^{2}\right)$	$\mathrm{tr}\left({\hat{\Phi }}^{-1}\right)$ $\left(m^{2}\right)$
Case A	5.4678	5.4620	/
Case B	5.5156	/	5.4620
Case C	2.5389	2.5310	/
Case D	2.5680	/	2.5310

**Table 4 entropy-23-01379-t004:** MAP estimation performances of three different methods with N=3,4,5,6.

Number	Method	MSE $\left(m^{2}\right)$	Bias Norm (m)
N=3	The proposed method	6.12	0.1472
	The method in [21]	12.35	0.8225
	The method in [22]	14.67	1.3557
N=4	The proposed method	4.32	0.0925
	The method in [21]	9.97	0.4634
	The method in [22]	11.43	0.8143
N=5	The proposed method	1.54	0.055
	The method in [21]	4.81	0.2415
	The method in [22]	5.94	0.5468
N=6	The proposed method	0.48	0.0123
	The method in [21]	1.61	0.1022
	The method in [22]	2.58	0.3967

## Data Availability

Not applicable.

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
