# Peer review of "Optimal 3D Angle of Arrival Sensor Placement with Gaussian Priors"

_entropy, 2021, doi:10.3390/e23111379_

Round 1

Reviewer 1 Report

The paper discusses an interesting problem of optimizing sensor locations for detecting an uncertain target. However, what the reviewer concerns is though the authors have mentioned some existing works/methods, there is no statement to declare what contributions are contributed in the current paper. Is the proposed approach better than the existing ones? In other words, it is expected to demonstrate improvement of the proposed method as compared with the existing ones at least in the results section.

Minor issues: Some abbreviation must be explained.

  • What do WIV and IV in lines 25 and 26, page 1 mean?
  • Row to row in matrix J2 in (9) should be presented clearly.
  • What does ML in page 7 mean?
  • \phi^{r, -1} in (35) and likewise should be represented to avoid confusion r-1.

Reviewer 2 Report

This paper presents the sensor placement optimization problem in three-dimensional (3-D) space for Angle of Arrival (AOA) target localization by using Gaussian priors.

My comments can be summarised as follows:

  1. In the "Abstract", the quantitative performance achievement of the proposed method should be given here.
  2. It is not clear the novelty and contribution of this work. Perhaps authors should provide a table to compare the performance, accuracy, speed, number of sensors, etc this proposed method with other methods available in the literature.
  3. Figure 3 shows the proposed Resistor network model for optimal sensor orientation for one sensor. It is not clear how this was proposed and linked to eqs. (38), (39) and (40).
  4. Only four configurations were proposed in Figure 3 and Figure 4. why ?? Can it be other configurations ??
  5. How many configurations for the resistor model for N>3 ?
  6. In Simulation Studies, how authors come out with the pre-defined values, i.e. Po, d, initial sensor location etc ?
  7. Authors used ref.[21], in their simulated model in section 5.1. What are the differences between this work and ref. [21] ?
  8. In Figure 5 and 6, what are the colours implication on the plots ?? Can authors provide the color bar ? More explanations are required for these figures.
  9. What are the radio propagation characteristics, i.e. transmitted power, RSSI level, antenna types, propagation environment etc were considered in this paper ??
  10. How the proposed method can be implemented in the practical experiment ?? Any experimental data/results to verify the proposed method ?? This will strengthen the quality and credibility  of this work.
  11. Some typo mistakes were found as follows. Please correct them. (a)Page 3, line 93, Please check the typo mistake, "taregt" should be "target"   (b) Page 2, line 82, "Extensive simulation studies are present to demonstrate the analytical findings" should be revised "Extensive simulation studies are presented to demonstrate the analytical findings".

Round 2

Reviewer 1 Report

Thanks the authors for addressing the reviewer's concerns in the revised version. The current manuscript can be considered for publication.

Author Response

Thank you very much for your help.

Reviewer 2 Report

After the first round of review, although authors have addressed some of my comments. But, I do feel many other comments not being answered to my satisfaction. Hence, my decision remains as it is.  In general, my comments can be summarised as follows:

  1. In the "Abstract", the quantitative performance achievement of the proposed method should be given here. Authors failed to provide this and reader cannot understand what have been done in this work without this.
  2.  It is not clear the novelty and contribution of this work. Perhaps authors should provide a table to compare the performance, accuracy, speed, number of sensors, etc this proposed method with other methods available in the literature. I cannot see any new table has been provided here to clearly demonstrate the significant of the work.
  3. Authors only compared the performance of their proposed method with Ref. [21]. How about others ??
  4. Authors assume there are no multipath Effects, doppler Spread, and propagation delay in the  propagation environment. This is only for ideal scenario and this never be the case for practical environment. 
  5. Authors also mentioned that the antenna is assumed to be a directional antenna or antenna array on each sensor. Again, if authors are considering a moving sensor, how this is realistic to have directional antenna. Perhaps omnidirectional antenna is more suitable ?
  6. In Figure 5, it shows the Optimal sensor orientation for one sensor. The given color bar should be all using the same color scale for better comparison. This also applies to Figure 6 and Figure 7.
  7. How the proposed method can be implemented in the practical experiment ?? Any experimental data/results to verify the proposed method ?? This will strengthen the quality and credibility of this work. Authors not able to provide any data for this. However, can authors used other published experimental data for their work ?

Round 3

Reviewer 2 Report

Authors have addressed all my comments to my satisfaction. I have no further comments. Well-done.